# Comparative Genomic Hybridization (CGH) in New World Monkeys (Primates) Reveals the Distribution of Repetitive Sequences in Cebinae and Callitrichinae

**DOI:** 10.3390/biology14010022

**Published:** 2024-12-29

**Authors:** Vanessa Milioto, Vincenzo Arizza, Aiti Vizzini, Polina L. Perelman, Melody E. Roelke-Parker, Francesca Dumas

**Affiliations:** 1Department of “Scienze e Tecnologie Biologiche, Chimiche e Farmaceutiche (STEBICEF)”, University of Palermo, 90123 Palermo, Italy; vanessamilioto@gmail.com (V.M.); vincenzo.arizza@unipa.it (V.A.); aiti.vizzini@unipa.it (A.V.); 2Institute of Molecular and Cellular Biology, SB RAS, Novosibirsk 630090, Russia; polina.perelman@gmail.com; 3Laboratory Animal Sciences Program, Frederick National Laboratory for Cancer Research, Frederick, MD 21702, USA; melody.roelke-parker@nih.gov

**Keywords:** heterochromatin, pattern, dynamic, evolution

## Abstract

The Genomic Hybridization (CGH) of the total DNA from two individual animals labeled with two different fluorescent dyes are mapped on a target metaphase, which can be from the same species or from a different species. This approach permits identifying the pattern of distribution of repetitive sequences on chromosomes. Furthermore, it is possible to identify loss or gain of repetitive sequences among species through this approach, changes that can occur during the species evolution. These changes can be useful for delineating repetitive sequences dynamic or phylogenetic and conservation issues.

## 1. Introduction

The human genome includes coding regions recognized as euchromatin and highly repetitive sequences known as heterochromatin. Homologies and differences among genomes could be detected at different levels: at the chromosomal morphology level, syntenic association, insertion or deletion of certain DNA segments or genes, and heterochromatin distribution. Initially, heterochromatin sequences were detected as constituents of peaks in cesium chloride density gradient bands and differentiated from the remaining genomic DNA by their A/T content [1]. Then, these sequences were recognized as satellite DNA, consisting of tandemly arranged repeats, at first seen as serving no useful purpose and known as the dark matter of the genome [2,3]. These satellite DNA are known as tandemly repetitive DNA and have now been associated with genome function [3,4,5,6,7]. It was hypothesized that these sequences play an important role in chromosome organization and structure stabilization, the recognition and segregation of chromosomes in mitosis and meiosis, and gene activity regulation [3]. Furthermore, heterochromatin was pointed out as one of the putative responsible causes of karyological diversification in several vertebrate models [8]. Indeed, it was already hypothesized that changes in the amount and distribution of repetitive DNA sequences can affect chromosome evolution and speciation even if their role is not well understood [4,5]. For all these reasons mentioned, the analysis of repetitive sequence compositions and localization in different genomes [9,10,11,12,13] is particularly useful in studies of genomic evolution. Furthermore, tandemly repetitive DNAs have high intraspecific sequence homogeneity and interspecific differences, making tandemly repetitive DNAs potential taxonomic markers and, in some cases, allowing their use for phylogenetic inference [13,14,15,16]. The identification of homologies from apomorphisms such as species-specific characters in genomes of closely related species defines genetic uniqueness [16,17,18,19,20]. A large portion of the genome in Primates consists of repetitive DNA sequences, including tandem and dispersed satellite repeats [3]. It was hypothesized that they evolve by gene conversion mechanisms and unequal crossing-over known as a concerted evolution [20].

Among Platyrrhini, different Cebidae species were studied first by southern blot and patterns of enzyme digestion showing that several types of heterochromatin arrangement are present in the group [21,22,23]. The capuchin monkey species analyzed here, from the Cebinae subfamily, were previously considered to belong to the same genus *Cebus.* Later, the genus was divided into two genera, one with the gracile (or untufted) *Cebus* and the second with the *Sapajus,* robust (or tufted) capuchins [24].

The Capuchin species *Cebus capucinus* and *Sapajus apella* (previously *Cebus apella*) have the same diploid number 2n = 54 and the same chromosome complements, but they differ through inversions and heterochromatin distribution, as previously revealed by C-banding and the mapping of specific repetitive sequence probes [15,21,22,23,24,25,26,27,28,29,30,31,32,33,34,35,36,37].

The other two species considered in this study are from the Callitrichinae subfamily and were first recognized as tamarins from the same genus *Saguinus.* This genus was then divided into two genera: *Saguinus* with all “large” species and *Leontocebus* with the smallest species [38]. The tamarin species *Saguinus mystax* and *Leontocebus fuscicollis* share the same diploid number 2n = 46 and rather similar G-banded karyotypes with similar chromosome morphologies but differ through paracentric or pericentric inversions and differences in heterochromatin distribution, as previously revealed by a C-banding and the mapping of specific repetitive sequence probes [25,26,39,40,41,42,43,44,45].

In this work, we analyzed the genomes of *Cebus capucinus*, *Sapajus apella*, *Saguinus mystax,* and *Leontocebus fuscicollis* (Cebidae) using intra- and interspecies Comparative Genomic Hybridization (iCGH) to detect balanced or unbalanced heterochromatin DNA distribution patterns between the genomes.

Apart from species-specific hybridization in *Sapajus apella, Leontocebus fuscicollis,* and *Sapajus mystax*, we performed interspecies CGH also known as cross-species CGH because it is important to understand the dynamic of heterochromatin occurring during evolution in phylogenetically close species with equal or different diploid numbers.

*Cebus* species have a high tandemly repetitive DNA amplification in genomes as CGH results already showed in some previously analyzed species [33] in agreement with classic cytogenetics. C-banding data suggest that species can be distinguished by the non-centromeric heterochromatin blocks on some chromosomes with a different chromosomal position [22,33,34,35,36,37,38]. Both classical cytogenetic methods and CGH were performed to detect heterochromatin in capuchins, while CGH studies on tamarins’ chromosomes were performed here for the first time. So far, a lot of cytogenetic information was gathered on *Cebus* and *Saguinus,* while data are poor for the recently recognized genera *Sapajus* and *Leontocebus*.

## 2. Materials and Methods

### 2.1. Cell Culture and Chromosome Preparations

For the species *Cebus capucinus* (CCP), *Leontocebus fuscicollis* (LFU)*,* and *Saguinus mystax* (SMY), fibroblast cell cultures were prepared with DMEM cell culture media (Dulbecco’s Modified Eagle’s medium, Sigma-Aldrich, St. Louis, MO, USA), supplemented with 20% fetal bovine serum (GIBCO/Thermo Fisher Scientific, Waltham, MA, USA), 1% penicillin-streptomycin (Sigma-Aldrich, St. Louis, MO, USA) and incubated at 37° [36].

*Sapajus apella* metaphases were prepared from phytohemagglutinin stimulated lymphocytes in RPMI culture medium (GIBCO/Thermo Fisher Scientific, Waltham, MA, USA) with 1% penicillin (10,000 units/mL)–streptomycin (10,000 µg/mL) (Sigma-Aldrich, St. Louis, MO, USA). The metaphase chromosome spreads for CGH experiments were obtained according to standard procedures involving colcemid 0.01 ug/mL exposure (1 h, 37°), hypotonic treatment (0.075 M KCl, 15 min, 37 °C), and fixation with methanol:acetic acid (3:1) [10].

For detailed information regarding species samples used for CGH use Table 1 and Table 2.

### 2.2. DNA Isolation and Whole-Genome Amplification

From all cell cultures, total genomic DNA was extracted (QIAamp DNA Mini Kit, QIAGEN, Hilden, Germany) and amplified using a whole-genome amplification kit (GenomePlex WGA, Sigma-Aldrich, S.t Louis, MO, USA) according to the manufacturer’s protocol. Whole-genome amplification was performed since the concentration of DNA is crucial for the success of CGH experiments.

### 2.3. Probe Preparation

The total genomic DNA probe was labeled following a modified Nick Translation protocol in the presence of Cyanine 3 (Cy3)-dUTP for female specimens and 11-d-UTP fluorescein for male specimens.

Briefly, the amplified DNA was mixed with a polymerase buffer and DNAse I (0.01–0.03 U/µL) for DNA digestion. Incubation with DNA polymerase I was performed for 1 h and 30 min at 15°. The enzyme was inactivated at 0°.

### 2.4. Comparative Genomic Hybridization

Interspecific Comparative Genomic Hybridization (iCGH) with 500 ng of female (Cy3, red) and 500 ng of male (SG, green) probes was performed by fluorescence in situ hybridization (FISH) following previously described protocol [14,23] with some adjustments. The whole protocol was applied as it follows: probes were mixed, precipitated with ethanol, and suspended in a hybridization mixture containing 50% formamide, 20% dextran sulfate in 2× SSC, and Milli-Q sterile water. The mixture was denatured at 80 °C for 8 min. Slides with chromosomal spreads were denatured in 70% formamide in 2× SSC, at 70 °C for 1 min, followed by dehydration in an ethanol series (70%, 90%, and 100% sequentially). Hybridization was conducted in a wet chamber at 37° for 48 h. Post-hybridization washes followed standard protocols with low stringency conditions: at 42 °C with 50% formamide followed by saline sodium citrate 2× SSC and 4× SSC (0.1% Tween-20) buffer solutions. The chromosomes were stained with 4′,6-diamidino-2-phenylindole (DAPI). This procedure was conducted on metaphase spreads as reported in Table 2.

### 2.5. Image Acquisition and Processing

The metaphases were analyzed under a Zeiss Axio2 epifluorescence microscope and captured using a coupled Zeiss digital camera. At least 10 metaphase spreads were analyzed for each CGH experiment. Specifically, three images per metaphase were obtained: DAPI, Fluorescein, and Cy3. Image processing was performed using Photoshop CS (Adobe).

The ideogram of *Cebus/Sapajus* and *Saguinus/Leontocebus* reporting heterochromatic loci identified by C banding, in agreement with literature data, are reported in Figure 1 [23,25,26,45].

**Figure 1 biology-14-00022-f001:**
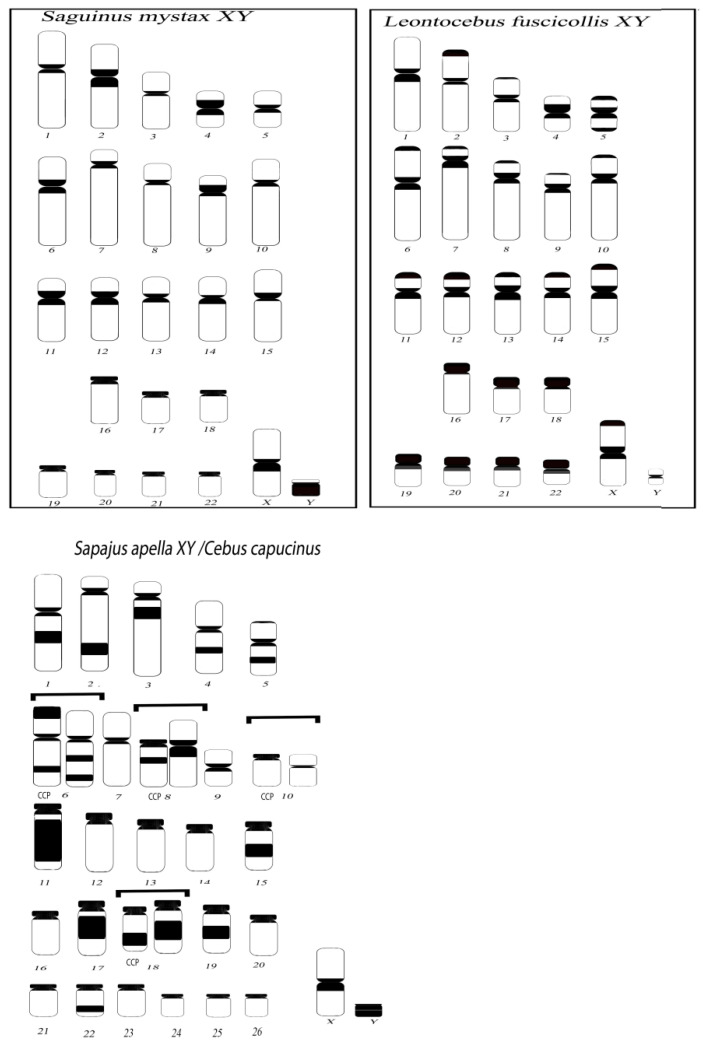
C bands on *Saguinus mystax, Leontocebus fuscicollis, Sapajus paella*, and *Cebus capucinus* haploid set of chromosomes which number is reported below each form. Last two species (SAP, CCP) have the same pattern of heterochromatin loci distribution except for chromosomes with different morphology, which are reported on square (CCP forms are at the left side) as previously shown [23].

**Table 2 biology-14-00022-t002:** The experimental setup for CGH: each hybridization has always been performed between a female specimen genomic DNA (♀, labeled in red fluorescent dye) versus (x) a male specimen genomic DNA (♂, labeled in green fluorescent dye) on target species metaphases (acronyms and figure numbers are also reported).

CGH of DNA from Species-1 on Target Species-2 Metaphases	Target Species (Metaphases)	Experimental Setup and Designations	Figure
*Sapajus apella* versus *Sapajus apella*	on *S. apella*	SAPfxSAP13m on SAPf ♀	Figure 2a
*Sapajus apella* versus *Cebus capucinus*	on *C. capucinus*	SAPfxCCP5m on CCP5m ♂	Figure 2b
	on *S. apella*	SAPfxCCP5m on SAPf ♀	Figure 2c
*Leontocebus fuscicollis* versus *Leontocebus fuscicollis*	on *L. fuscicollis*	LFU4fx LFU5m on LFU5m ♂	Figure 2d
		LFU4fxLFU5m on LFU4f ♀	Figure 2e
*Saguinus mystax* versus *Saguinus mystax*	on *S. mystax*	SMY1fxSMY4m on SMY4m ♂	Figure 2f
		SMY1fxSMY4m on SMY1f ♀	Figure 3a
*Saguinus mystax* versus *Leontocebus fuscicollis*	on *L. fuscicollis*	SMY1fxLFU5m on LFU5m ♂	Figure 3b
*Sapajus apella* versus *Leontocebus fuscicollis*	on *Saguinus mystax*	SAPfxLFU5m on SMY1f ♀	Figure 3c
		SAPfxLFU5m on SMY4m ♂	Figure 3d
*Sapajus apella* versus *Saguinus mystax*	on *L. fuscicollis*	SAPfxSMY4m on LFU4f ♀	Figure 3e
*Leontocebus fuscicollis* versus *Saguinus mystax*	on *S. apella*	LFU4fxSMY4m on SAPf ♀	Figure 3f

**Figure 2 biology-14-00022-f002:**
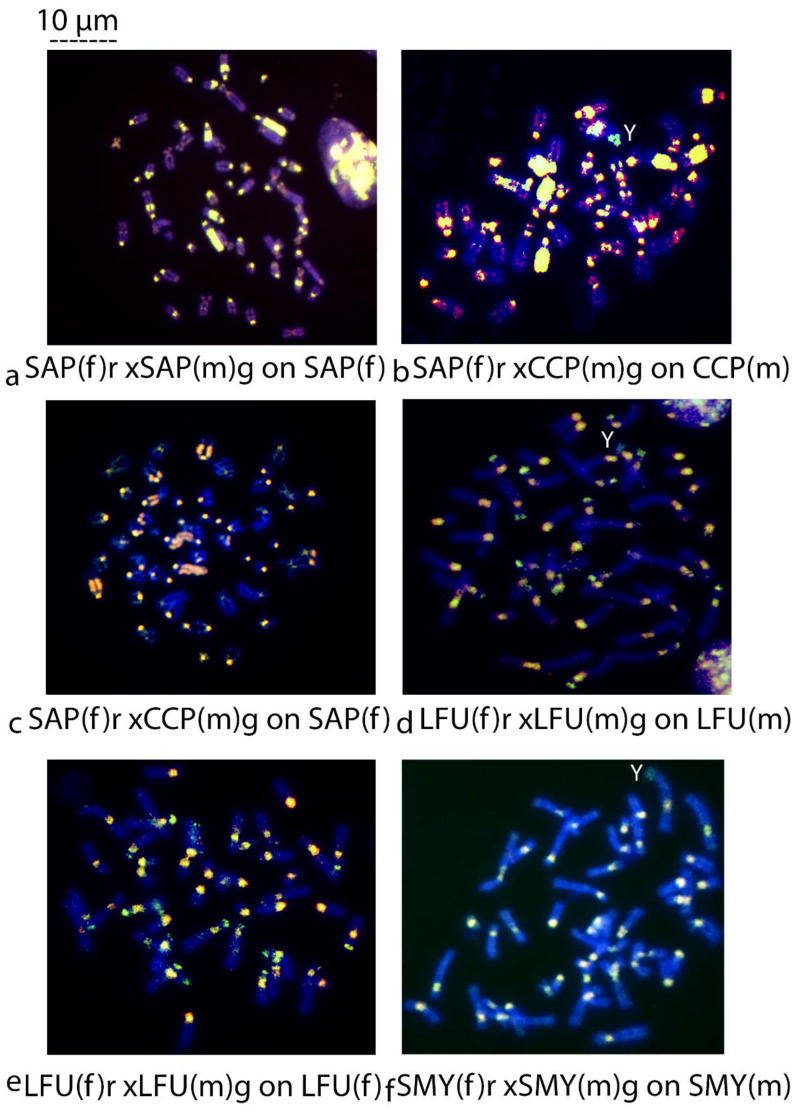
(**a**–**f**) Comparative Genomic Hybridization (CGH). (**a**) SAP(f) red xSAP(m) green on SAP(f); (**b**) SAP(f) red xCCP(m) green on CCP(m); (**c**) SAP(f) red xCCP(m) green on SAP(f); (**d**) LFU(f) red xLFU(m) green on LFU(m); (**e**) LFU(f) red xLFU(m) green on LFU(f); (**f**) SMY(f) red xSMY(m) green on SMY(m). Scale bar 10 µm.

**Figure 3 biology-14-00022-f003:**
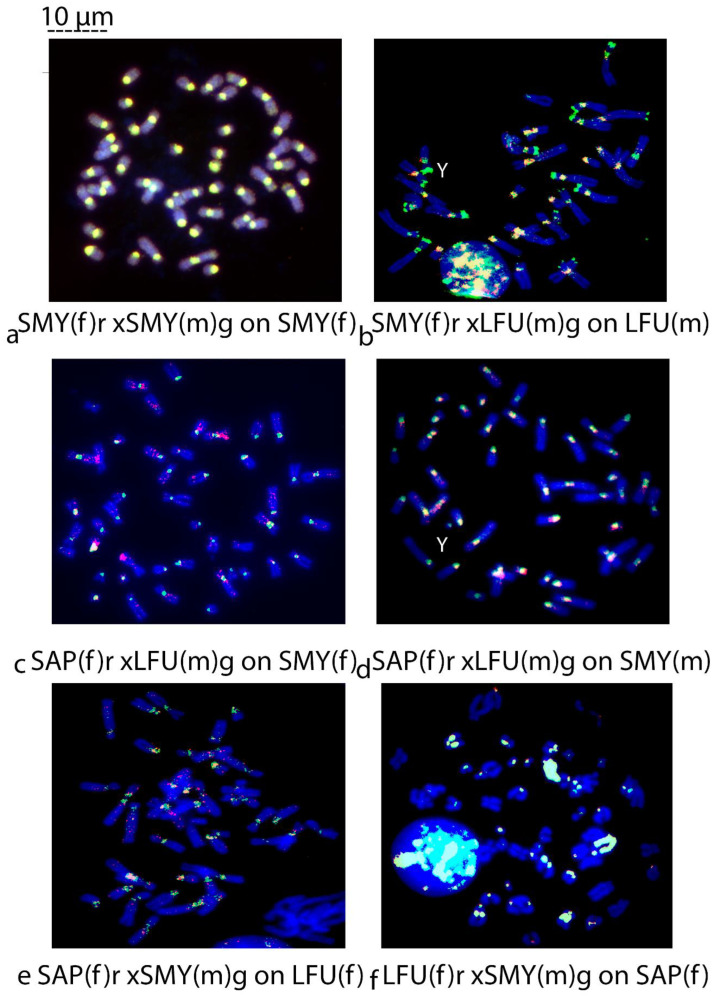
(**a**–**f**) Comparative Genomic Hybridization (CGH). (**a**) SMY(f) red xSMY(m) green on SMY(f); (**b**) SMY(f) red xLFU(m) green on LFU(m); (**c**) SAP(f) red xLFU(m) green on SMY(f); (**d**) SAP(f) red xLFU(m) green on SMY(m); (**e**) SAP(f) red xSMY(m) green on LFU(f); (**f**) LFU(f) red xSMY(m) green on SAP(f). Scale bar 10 µm.

## 3. Results

### 3.1. CGH in Cebidae

#### 3.1.1. Intra- and Interspecific CGH in Capuchin Monkeys

CGH of a female versus a male mapped on female metaphases of *Sapajus apella* shows interstitial blocks of bright balanced yellow signals covering a large part of several autosomal pairs and also signals at centromeres of most acrocentric chromosomes. Also, a bright yellow signal is seen at p-arm telomeres on one large submetacentric chromosome pair (Figure 2a).

The interspecies CGH of *S. apella* female versus *Cebus capucinus* male on *C. capucinus* male (Figure 2b) and on *S. apella* female (Figure 2c) metaphases show bright balanced interstitial signals in yellow, and at least two submetacentric chromosomes have signals at the centromeric position. Furthermore, on *C. capucinus* male metaphases, almost the entire Y acrocentric chromosome is green (Figure 2b). On centromeres of many chromosome pairs, red signals are also present in both *Sapajus apella* (Figure 2a) and *Cebus capucinus* (Figure 2b)

The interspecies CGH of *S. apella* female versus *C. capucinus* male on *S. apella* female metaphases (Figure 2c) shows the same blocks of balanced yellow interstitial heterochromatin seen (Figure 2a).

#### 3.1.2. Intra- and Interspecific CGH in Tamarins

The intraspecific CGH of female versus male on *Leontocebus fuscicollis* male (Figure 2d) and female (Figure 2e) metaphases show very big, amplified balanced yellow centromeric signals in both males and females. Interestingly, not balanced telomeric green signals are shown, with a male DNA origin on both male and female samples (Figure 2d,e). In male *L. fuscicollis*, the sex chromosome is a small biarmed Y chromosome covered by a green signal (Figure 2d).

The intraspecific CGH of female versus male on *Saguinus mystax* male (Figure 2f) and female metaphases (Figure 3a) in both samples shows uniformly balanced centromeric yellow signals and lighter yellow telomeric areas on some chromosomes. The Y chromosome is a small acrocentric chromosome covered by a green signal (Figure 2f).

The interspecific CGH of *S. mystax* female versus *L. fuscicollis* male on *L. fuscicollis* male (Figure 3b) shows slightly less extensive balanced signals in yellow at the centromeric position and not balanced, big, and amplified telomeric signals in green on many submetacentric chromosomes and on the small biarmed Y chromosome which is almost all green.

#### 3.1.3. Interspecific Reciprocal CGH Capuchin Monkeys Versus Tamarins

The interspecific CGH of *S. apella* female versus *L. fuscicollis* male on *S. mystax* female (Figure 3c) and male (Figure 3d) metaphases show unbalanced interspersed red signals of *S. apella* origin along the chromosome arms. Some unbalanced red signals are also at the telomeric position of some submetacentric chromosome pairs (Figure 3c), and the green signal is on the Y chromosome (Figure 2d). Furthermore, the centromeric regions of submetacentric chromosome pairs in both the male and the female show balanced yellow signals and unbalanced green signals of *L. fuscicollis* origin (Figure 3c,d). The interspecific CGH of *S. apella* female versus *S. mystax* male on *L. fuscicollis* (Figure 3e) shows few red signals of *S. apella* origin along the chromosome arms and at the centromeres balanced yellow signals and unbalanced green signals of *S. mystax* origin.

The reciprocal interspecific CGH of *L. fuscicollis* female versus *S. mystax* male on *S. apella* female (Figure 3f) shows balanced signals in yellow at centromere on some chromosome pairs, especially on chromosomes characterized by the big blocks of heterochromatin as previously shown in capuchin monkeys (Figure 2a–c). A few unbalanced centromeric signals are green on some chromosome pairs with *S. mystax* origin (Figure 3f).

### 3.2. Figures, Tables, and Schemes

As shown in Figure 2, the CGH probe was made of a combination of female (f) genomic DNA from sample 1 (labeled in red (r) fluorescent dye) with a male (m) genomic DNA from sample 2 (labeled in green (g) fluorescent dye) on the target species metaphases. It is worth noting that the first sample is female and always labeled in red, and the second is male always labeled in green. For intraspecific hybridization, samples 1 and 2 and the target species were from the same species. The target species could be from samples 1 or 2. In interspecific hybridization, samples 1 and 2 were different species. Target species could be from the same sample 1 or 2 (see Table 2 for the acronyms and the experimental setup). For example, Figure 2b shows an interspecific CGH of the probe composed of a combined genomic DNA from a female *S. apella* labeled in red (sample 1) and DNA from *C. capucinus* male labeled in green (sample 2) hybridized onto the chromosomes of the male *C. capucinus* (sample 2) = SAP(f) rxCCP(m) g on CCP(m). The Y chromosome is indicated with a white Y. The magnification is 1000×.

## 4. Discussion

Here, the pattern of distribution of the repetitive sequences in Cebidae was analyzed through iCGH, which allows the screening of the genome at chromosomal and subchromosomal levels in search of balanced or unbalanced DNA content, identifying changes as gains and losses in the heterochromatin content [46]. Those observations can also help to elucidate the dynamic of repetitive sequences’ evolution in species [33,47].

### 4.1. Intra- and Interspecific CGH in Capuchin Monkeys

The intraspecific CGH, *S. apella* versus *S. apella* on female *S. apella* metaphases (Figure 2a), shows big blocks in yellow in agreement with the extended blocks of constitutive heterochromatin described previously for other capuchin species by C bands (Figure 1), also through the mapping of specific repetitive probes such as the CAP-specific probe, while less correspondence is with signals of LINE-like probes [23,29,32,33]. Those heterochromatic interstitial blocks, indeed, overlap with a specific Platyrrhini repetitive sequence described for the first time in *S. apella* [21]. The CAP-specific sequence in Platyrrhini originates from a single-copy gene of Eutherian mammals that became repetitive through duplication and amplification [23]. The CAP-specific probe was mapped on capuchin monkeys [14] showing the interstitial block signal, but it does not hybridize to all centromeres. Most of the LINE sequences previously mapped in those species are instead not in correspondence with blocks of constitutive heterochromatin. This evidence is not surprising since certain repetitive sequences at a specific chromosomal locus are not necessary in correspondence with heterochromatic C bands obtained by C-banding or immunofluorescence. Repetitive sequences, indeed, can occur both within and outside heterochromatin; consequently, not all repetitive sequences are heterochromatic loci [6,7]. In support of this perspective, there are example of genomes with abundant repetitive sequences and scarce heterochromatin [29,43].

The interspecies CGH *S. apella* versus *C. capucinus* on *C. capucinus* male (Figure 2b) showed almost the same pattern of hybridization in yellow as seen in *S. apella* (Figure 2a,c), highlighting the extended interstitial blocks of the heterochromatin and also showing yellow signals at the centromere of two submetacentric chromosome pairs. Those CGH results distinguish *S. apella* from *C. capucinus*: the yellow signals are present at centromeres in the latter species and absent in the former. Those differences are presumably due to inversions that changed the interstitial signals to a centromeric position as previously shown in other species of capuchin monkeys through the mapping of other specific repetitive probes [23,33,36,37]. In particular, those inversions were previously hypothesized through the pattern of the LINE-like probes previously mapped in the two species in the analysis [23]. Regarding the unbalanced heterochromatic DNA, the green signal on the Y chromosome in C*. capucinus* male (Figure 2b) could be explained by Y-chromosome- repeats that are under-represented in the genome of the corresponding *S. apella* female. Furthermore, the bright red CGH signals at centromeres from *S. apella* female in *C. capucinus* and *S. apella* demonstrate a slight species-level divergence of chromosome repeats among *Sapajus* and *Cebus* amplified in the former in agreement with the time of radiation estimated to be 2.91 Mya and 2.65 Mya for the latter (Figure 2b) [48].The interspecies hybridization between *S. apella* versus *C. capucinus* on *S. apella* female metaphases (Figure 2c) confirms the pattern of hybridization already seen (large interstitial blocks and centromeres of acrocentric chromosomes in orange presumably because more amplification occurred in *S. apella*) (Figure 2a,b). These results, reported for the two capuchin species analyzed here, also agree with previous CGH literature data. CGH between other capuchin species (*C. libidinosus* and *C. nigritus* DNA hybridized on *C. libidinosus*) led researchers to show differences in heterochromatin distribution between the two species [36]. Also, previous CGH of *C. paraguayanus* and *C. nigritus* (*Sapajus nigritus*) [33] showed that the species can easily be distinguished at the chromosomal level due to the absence or the presence of non-centromeric repetitive sequence blocks on some chromosomes in agreement with classical cytogenetic comparisons [30,31].

### 4.2. Intra- and Interspecies CGH in Tamarins

The intraspecies CGH of female versus male on *L. fuscicollis* male (Figure 2d) and female (Figure 2e) metaphases show big balanced yellow signals on both samples at centromeres and big unbalanced green signals at telomeric regions of many submetacentric chromosomes, in agreement with C bands (Figure 1) with a predominant presence of the male DNA relative to the female counterpart in both males and females. The Y chromosome is a peculiar small biarmed chromosome covered by an extensive green signal, of which its composition needs to be verified.

The intraspecies CGH of female versus male on *Saguinus mystax* male (Figure 2f) and female metaphases (Figure 3a) showed in both samples balanced centromeric signals in yellow and no signals at telomeric positions despite the phylogenetic proximity *with L. fuscicollis* [49].

Indeed, the interspecies CGH of *S. mystax* female versus *L. fuscicollis* male on *L. fuscicollis* male (Figure 3b) showed small and less amplified but balanced signals in yellow at the centromeres and large unbalanced amplified telomeric signals of *L. fuscicollis* origin in green. This hybridization shows telomeric heterochromatin blocks which can be explained as an apomorphic feature found in *L. fuscicollis* in agreement with previous C-banding data [23,33,45]. The presence of these large telomeric signals could be a result of repeat amplification/transfer in *L. fuscicollis* or of an inversion occurring during evolution, involving centromeric sequences, as was previously hypothesized [23]. This inversion could have involved the centromeric heterochromatin that is indeed less extended in this species. Furthermore, it would be useful to verify if the Y chromosome green signal (Figure 2d) composition is of the same repetitive sequence as the telomeric blocks found on autosomes. Previously, C-banding revealed heterochromatin at the telomeric position also in *Saimiri sciureus* (Saimiriinae, Cebidae) [50]; thus, the presence of amplified repetitive sequences in telomeric areas might indicate that these areas are prone to developing large blocks of heterochromatin in different branches of Cebidae, and this feature can be the result of convergent evolution in the two subfamilies. Further analysis with telomeric sequence probes is needed to see if these telomeric blocks are the amplified telomeric repeats or a result of the expansion of some other repeat at the centromere followed by inversion and resolving the doubt about homoplasy or apomorphism.

### 4.3. Interspecies Reciprocal CGH Between Capuchin Monkeys and Tamarins

The interspecies hybridization between *S. apella* versus *L. fuscicollis* on *S. mystax* female (Figure 3c) and male metaphases (Figure 3d) shows in both samples few unbalanced red interspersed signal of *S. apella* origin on some chromosomes in euchromatic regions and some light signals at the telomeric regions of submetacentric chromosomes on *S. mystax*. This less amplifies interspersed repetitive sequences can be explained as no constitutive heretochromatin, which is in agreement with previous hypothesis [6,7]. The CGH indicates, on the other hand, the lack of green telomeric signals of a *L. fuscicollis* origin in *S. mystax*. At the centromeric position, the presence of yellow balanced and green and red unbalanced heterochromatin indicates the presence of shared repeats (yellow) present in *S. apella* and *L. fuscicollis*, unbalanced (green) repeats shared by *L. fuscicollis* and *S. mystax* and a few unbalanced (red) repeats shared by *S. apella* and *S. mystax*. Other centromeres are green and represent tamarin-specific heterochromatin presumably due to amplification and concerted evolution in the species. The Y acrocentric chromosome shows green heterochromatin at centromeric position (Figure 3d) and presumably chromosome repetitive sequences are the same in *S. mystax* and *L. fuscicollis* (Figure 3b,d). However, in *L. fuscicollis, the* Y chromosome is a little biarmed chromosome green covered, while in *S. mystax*, it is an acrocentric chromosome and presumably the different morphology is due to an inversion or alternatively to a simple signal amplification, as above hypothesized.

The interspecies CGH of *S. apella* versus *S. mystax* on *L. fuscicollis* female (Figure 3e) hybridization shows big yellow balanced signals at centromeres next to some unbalanced less extended (green) signals in common with *S. mystax* or (red) in common with *S. apella*. The presence of unbalanced green or red signals at centromeric areas in both *S. mystax* and *L. fuscicollis* (less extended in *L. fuscicollis*, in red, Figure 3e) can presumably arise due to the mechanism of the amplification of repeated sequences (Figure 3c–e), then conserved and shared between one or more species. In brief, the two described cross-species CGH of *S. apella* versus S*. mystax* on *L. fuscicollis* and of *S. apella* versus *L. fuscicollis* on *S. mystax* show balanced heterochromatin region at centromeres shared between *Sapajus* and the two tamarin species. There are unbalanced green or little red signals at centromeres close to these balanced yellow signals that indicate heterochromatin of respectively of tamarin and cebidae origin in both S*. mystax* and *L. fuscicollis*, being more or less extended in the former or in the latter species. These heterochromatin sequences in the two tamarin species presumably arose due to amplification or through concerted evolution mechanisms. If these green-highlighted centromeric sequences in both tamarin species were the same sequence, it would be a synapomorphic derived feature for tamarins. Further analysis with other techniques is needed to investigate the origin of these underlined sequences. Furthermore, the results confirm that no yellow or green signals are at telomeric positions (Figure 3e), confirming that this repetitive sequence is not in *S. apella* or *S. mystax* genomes.

The cross-species reciprocal CGH of tamarins, of *L. fuscicollis* versus *S. mystax* on *S. apella* female (Figure 3f) shows balanced interstitial blocks in yellow along the arms of submetacentric chromosome pairs, with the same pattern in capuchin monkeys confirming that this big amount of heterochromatin is a common feature in Cebinae and Callitrichinae. Some unbalanced centromeric sequences in green are on a few chromosome pairs. The presence of these green signals of *S. mystax* origin at centromeric positions of *S. apella* acrocentric chromosomes (presumably not gained or lost in *L. fuscicollis*) could be the same as the repetitive sequences of *S. apella* stained in red, which are seen as intercalated and particularly amplified in *S. mystax* (Figure 3c,d). If they were the same sequences, they would be a link between Cebinae and Tamarinae. These repetitive sequences have to be better investigated with other techniques such as sequencing to explain their origin and to verify if it could be a phylogenetic marker linking the two species.

## 5. Conclusions

The species pairs studied here have the same diploid number, Cebinae 2n = 54 and Callitrichinae 2n = 46. From an evolutionary perspective, the capuchin monkeys (*Sapajus* and *Cebus*) from Cebinae have conserved karyotypes close to the hypothetical ancestral New World monkeys’ karyotype [21], while tamarins from (*Saguinus* and *Leontocebus*) Callitrichinae have more derived karyotypes.

iCGH highlights the repetitive sequences pattern within Cebinae species showing the presence of large blocks of interstitial heterochromatin in capuchin monkeys and same variations at both centromeric and non-centromeric regions. The results led to a hypothesis that the pattern seen in Cebinae could have been the ancestral distribution and that during evolution this genomic diversification could occur by heterochromatin expansion and amplification and rearrangements in Cebinae. Indeed, between the closely related genera *Sapajus* and *Cebus*, the diversification has likely occurred through amplification and inversions in agreement with previous hybridizations of other repetitive probes in Capuchin’s species [15,24,33,36,37].

The iCGH within the Callitrichinae species analyzed shows the presence of large centromeric heterochromatin balanced in both *L. fuscicollis* and *S. mystax* and unbalanced heterochromatin at telomeric positions in *L. fuscicollis* in agreement with previous C banding [45].

The reciprocal cross species CGH permitted to show the same content of repetitive sequences with a different pattern of distribution between the two species groups studied. This evidence let us to hypothesize that during evolution from Cebinae, where repetitive sequences were in blocks, to the more derived Callithricinae forms, these sequences were differently organized possibly through a variety of mechanisms; in particular, rearrangements, amplification, concerted evolution, and repetitive were especially found at centromeres. Some of these sequences highlighted in this study are shared by *Sapajus, Saguinus*, and *Leontocebus* species and could be seen as symplesiomorphies, and other sequences can be shared derived synapomorphies; nevertheless, other sequences can be automorphisms such as the big telomeric on in *L. fuscicollis*. This hypothesis agrees with a few molecular studies that recently allowed researchers to overcome the classic difficulties of sequencing repetitive sequences. This study permitted researchers to sequence and distinguish conserved, derived, and species-specific tandem repetitive sequences in New World monkeys [51]. Additional analysis of these repetitive sequences is needed to uncover the possible usefulness of phylogenetic markers and their genome role and function.

## Figures and Tables

**Table 1 biology-14-00022-t001:** List of the analyzed species, name acronym, cell type, and provenience.

Family	Latin Name/Common Name, 2n	Acronym	Cell Type
Cebidae Subfamily Callitrichinae	*Saguinus mystax*/Moustached Tamarin, 2n = 46	SMY	Fibroblast cell line
	*Leontocebus fuscicollis*/Saddleback Tamarin, 2n = 46	LFU	Fibroblast cell line
Subfamily Cebinae	*Cebus capucinus*/White-faced Capuchin, 2n = 54	CCP	Fibroblast cell line
	*Sapajus apella*/Brown Capuchin, 2n = 54	SAP	Lymphoblasts from blood

## Data Availability

The original contributions presented in this study are included in the article. Further inquiries can be directed to the corresponding author.

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
