# Peer review of "Comparative Genomic Hybridization (CGH) in New World Monkeys (Primates) Reveals the Distribution of Repetitive Sequences in Cebinae and Callitrichinae"

_biology, 2024, doi:10.3390/biology14010022_

Round 1

Reviewer 1 Report

Comments and Suggestions for Authors

Vanessa Milioto and co-authors investigated the distribution of repetitive sequences in Cebinae and Callitrichinae using intra- and cross-species comparative genomic hybridisation (CGH) on mitotic metaphase chromosomes. Interspecies CGH revealed the dynamics of heterochromatic chromosomal regions during karyotype evolution in phylogenetically close apes. The data presented are convincing, the authors have provided all the necessary details of the experimental design for CGH and high quality micrographs.

Major comments:

Consider changing the term "Satellite DNA". The term "satellite DNA" is mainly historical. Now that complete T2T genome assemblies allow detailed characterisation of tandem repeat arrays, it is more appropriate to use the term "tandemly repetitive DNA".

Please consider including additional references here: “Recently satDNA has been associated with genome function. It has been hypothesized that these sequences play an important role in chromosome organization and structure stabilization, recognition and segregation of chromosomes in mitosis and meiosis, and gene activity regulation [3].”

“Interspecific Comparative Genomic Hybridization (iCGH) with 500 ng of female (Cy3, red) and 500 ng of male (SG, green) probes was performed by Fluorescence in situ hybridization (FISH) following previously described protocol [10, 19] with some adjustments.” – Please consider citing detailed step by step FISH protocols here.

Figures 1 and 2. Please indicate directly on the panels which fluorescent dye (red or green) was used to label each genomic DNA, and also indicate the sex of the labelled DNA used for CGH (e.g. on panel b - by making the text 'SAP(f)' red and 'CCP(m)' green). Please add a scale bar on each panel.

The 'Conclusions' section should be rewritten. It could be rephrased to avoid repeating information already provided in the 'Discussion' section, such as descriptions of the FISH results as 'yellow signals' or 'green signals'.

Minor comments:

Please rephrase the sentence “Furthermore, permit to map eventually loss or gain of …”

Please rephrase the sentence “Cebus species are perhaps the most interesting in this perspective due to their conspicuous chromosomal and genomic characteristics as CGH results have already shown in some previously analyzed species [29] in agreement with classic cytogenetics.”

Reviewer 2 Report

Comments and Suggestions for Authors

In this manuscript the authors use the intraspecies and interspecies Comparative Genomic Hybridization to study the distribution of repetitive sequences in 4 species that belong to the subfamilies Cebinae and Callitrichinae.

My observations, comments, and suggestions regarding the manuscript are outlined below:

Line 27-28: Repetitive sequences can be species-specific, but their preservation and conservation across closely related species are also common, as postulated by satDNA library hypothesis, and shown in numerous publications.

Line 28: Their role in speciation is not well understood; however, their role within the genome has been extensively studied, summarized in, for example: doi.org/10.3390/genes13071154 and doi.org/10.3390/genes11010072.

Line 55: The authors use only their references to convey the message that the analysis of repetitive sequence compositions and localization in different genomes is useful in studies of genomic evolution. I believe that at least few other references on this could/should be included. 

Please have in mind that the presence of a certain repetitive sequence at a specific chromosomal locus does not necessarily indicate that these loci are heterochromatic. Repetitive sequences can occur both within and outside heterochromatin (e.g. 10.3390/genes15040395, 10.3390/genes14030742), and there are examples of genomes with abundant repetitive sequences but extremely scarce heterochromatin. Please refer to “heterochromatin” throughout the manuscript only for chromosomal loci where its presence has been confirmed using established detection methods, such as C-banding or immunofluorescence against heterochromatin markers like H3K9me2/3.

Including an ideogram in the manuscript, placed before the hybridization results, would be very helpful. This ideogram should denote the positions of heterochromatic loci previously identified by C-banding on the chromosomes of the inspected species.

Table 1. I suggest that the “Acknowledgment” column is removed from the Table and the acknowledgments to be places in the appropriate section at the end of the manuscript.

Lines 150-154: unbalanced red signals are not mentioned. They are also omitted from the Discussion.

Lines 158-159: In addition to Y, there are also plenty other green unbalanced signals which are not mentioned. They are also omitted from the Discussion.

Line 267: How is it determined that the repeats are Y-specific?

Line 272: It would be useful if the divergence times of the species were shown, in addition to the mentioned phylogenetic proximity.

Line 303-307: There is also unbalanced red signal in Fig. 2e which is not being discussed.

The “Conclusion” section should be more focused and present only the major conclusions of the work. Big part of the current Conclusions belongs to the Discussion.

Minor observations:

Line 20: There is a typo in the word monkeys.

Line 150: Please italicize Sapajus apella.

line 192: Please italicize S. mystax.

Line 194: Acronym for Cebus capucinus brought in Table 1 is CAP, however, later throughout the manuscript CCP is used.

 Line 226: Please italicize S. apella.

Lines 235, 236: Please italicize C. capucinus.

Line 243: The reference is missing.

Line 262: Name should be abbreviated to L. fuscicollis.

Round 2

Reviewer 2 Report

Comments and Suggestions for Authors

The authors responded adequately to the comments and suggestions, therefore I recommend the publication of the manuscript in Biology.